# The Influence of Air Humidity on the Output Signal from an Ionization Smoke Detector in the Presence of Soot Nanoparticles

**DOI:** 10.3390/s22103639

**Published:** 2022-05-10

**Authors:** Tomasz Jankowski, Piotr Sobiech, Szymon Jakubiak

**Affiliations:** Central Institute for Labour Protection–National Research Institute (CIOP-PIB), Czerniakowska 16, 00-701 Warsaw, Poland; pisob@ciop.pl (P.S.); szjak@ciop.pl (S.J.)

**Keywords:** air quality, nanoaerosol, environmental monitoring, nanoaerosol exposure assessment

## Abstract

In 2019, the European Committee for Standardization (CEN) initiated work on the preparation of a strategy for air quality monitoring at workplaces. The aim was to determine the concentrations of nano-objects and their aggregates and agglomerates (NOAA) by means of direct measurements using low-cost sensors. There is a growing need for low-cost devices that can continuously monitor the concentrations of nanoparticles, and that can be installed where nanoparticles are used or created spontaneously. In search of such a device, in this study, a smoke detector with an ionization sensor was tested. The aim of the research was to investigate the response of the analog output signal with respect to changes in environmental parameters such as the relative humidity of air. The research was conducted in controlled laboratory conditions, and the results confirmed that an ionization detector could be used to measure the concentrations of nanoaerosols. The modified smoke detector detected soot particles smaller than 100 nm. The linear regression line was calculated for the relative humidity dataset and had a slope coefficient of −1.214 × 10^−4^; thus, the value of the output signal was constant during the experiment. The dependence on air temperature was approximated by a second-degree curve, with a slope coefficient of −8.113 × 10^−2^. Air humidity affected aerosol concentrations, which may be related to surface modification of nanoparticles.

## 1. Introduction

Nanomaterials have a wide range of applications in medicine, biotechnology, environmental protection, telecommunications, energetics, transportation, and different types of industries [1,2], and advancements in nanotechnology offer many applications and possibilities. However, exposure to nanomaterials may pose a risk to human health [3,4,5,6]. Nanoparticles or ultrafine particles can enter into the human lungs and alveolar area, which is the main route of exposure to nanoparticles [7,8,9], and subsequently enter the human blood circulation system or even cross the blood–brain barrier [10,11,12,13]. The definition of “nanoparticle” (NP) provided by the International Organization for Standardization (ISO) is “a nano-object with three external nanoscale dimensions”, where nanoscale is defined as a size range from approximately 1 to 100 nm [14,15]. The term “ultrafine particles” (UFPs) is used when referring to nanomaterials that are highly related to anthropogenic emission sources or natural emission sources. UFPs are a fraction of ambient particulate matter (PM 0.1) [16,17]. For convenience, in this article, the term “nanoparticles” is used for both airborne NPs and UFPs.

Workers employed in the nanotechnology sector are most often exposed to nanoparticles, followed by academics and people employed in research institutes, and finally, consumers and users of nanoproducts. Quantitative information on nanoparticle concentrations is important for risk management of the surrounding environments [18,19,20].

Currently, exposure assessments for microparticles are based on measuring mass concentrations with a gravimetric analysis (weighing the filter before and after sampling the particle mass of a collected sample) [21,22,23]. Because of the minor impact of nanoparticle mass on the overall mass concentration of aerosols, they are difficult to assess using this approach [5]. Therefore, NP emissions are usually characterized by particle number concentration (particles/cm^3^) [20,24,25,26,27,28]. The instruments most commonly used for nanoparticle measurements are a condensation particle counter (CPC), a diffusion size classifier (DSC), a differential mobility analyzer (DMA), a scanning mobility particle sizer (SMPS), and an electrical low-pressure impactor (ELPI) [29,30,31]. A CPC is based on nanoparticle condensation on liquid vapors that are enlarged to a size measurable by an optical detector, a DSC is based on electrical charging of aerosols, a DMA classifies charged nanoparticles basing on their mobility in an electric field, a SMPS measures particle size distribution in relation to their electrical mobility diameter with electrometers and is usually combined with CPC, and an ELPI device is equipped with a cascade impactor and particles are directed on the proper level depending on their charge and then counted in relation to the aerodynamic diameter. However, the high cost of these devices is a disadvantage. Therefore, low-cost particulate matter sensors have gained a lot of attention by making applications feasible that are prohibitively expensive using traditional, laboratory-grade devices [32,33,34].

There is a growing need for low-cost devices for continuous monitoring of nanoparticle concentrations, which can be installed where such particles are used or created spontaneously. The research described in this paper is in line with the global trend of searching for new, inexpensive methods for measuring nanoparticle concentrations in the air [32,33,34,35,36,37,38]. The utilizing of ionization chambers used in smoke detectors to measure the concentration of submicrometric particles has already been proposed in several research papers [35,36,37,38]. Litton et al. [35,36] applied both optical and ionization smoke detectors for concentration measurements of micrometer and submicrometer aerosols. Later, this idea was developed by Edwards et al. [37], who showed that an ionization sensor was approximately five times more sensitive to the presence of fine particles, while a photoelectric sensor was about five times more sensitive to the presence of coarse particles. The lower detection limit for fine particles has been estimated to be 17 µg m^−^^3^. Moreover, it has been reported that, in addition to particle concentration, environmental conditions such as temperature, humidity, and pressure also influence the measured output signal value. Dahl et al. [38] proposed an application using a modified ionization smoke detector as a low-cost nanoparticle monitor. To overcome the influence of environmental conditions on the output signal, measurements were conducted in filtered and untreated air. The lower detection limit for 100 nm particles was estimated to be 15,000 particles/cm^−^^3^. The tested sensor showed a linear response for concentration changes in KCl calibration aerosol, candle smoke, and welding fumes.

In this study, we present the results of research on combining an optical particle counter with a detector as a low-cost measuring device for continuous monitoring of aerosol concentrations over a wide range of particle sizes. A smoke detector with an ionization sensor was tested as a low-cost device for monitoring nanoaerosol concentrations. The aim of the research was to investigate changes in the value of the analog output signal in response to changes in environmental parameters such as the relative humidity of air. The research was conducted in controlled laboratory conditions. Increased air humidity may affect both the process of generating and transporting ions in the detector, as well as interact with the surface of the particles.

## 2. Materials and Methods

### 2.1. Nanoparticle Detector

In this study, we propose the use of a smoke detector with an ionization sensor as a low-cost device for monitoring nanoaerosol concentrations in air. Figure 1 shows the model nanoparticle detector. Figure 2 presents a block diagram of the proposed nanoparticles detector. The basic operating principle was to compare the output signal values from the ionization chamber of the sensor.

A smoke detector DIO-40 (Polon-Alfa, Bydgoszcz, Poland) [39] was modified by the manufacturer by exposing two voltage signals: the supply voltage and the signal from the electrometer, which measures the electric potential of the floating electrode. Although the smoke detector, for its primary application of fire sensing, was equipped with a system to neglect the influence of environmental conditions such as air temperature and air humidity, it was bypassed in this experiment. The difference between both analog signals was measured using an electronic circuit in two stages. In the first stage, both signals were amplified using a LM358 operational amplifier (Texas Instruments, Dallas, TX, USA) with a gain equal to 1. In the second stage, the LM358 operational amplifier was used to determine the difference between both signals (the supply voltage was connected to the positive side, while the signal from the electrometer was connected to the negative side). The LM358 operational amplifier was used as a voltage comparator for the two voltages supplied to its input and, depending on which input voltage was lower and which was higher, it exposed a logical state of low or high to the output. The built-in frequency compensation increased the operational stability of the LM358. The resulting signal was measured using a built-in analog-to-digital converter (ADC) in the Microchip SAMD21 microcontroller (Microchip Technology Inc., Chandler, AZ, USA) installed on the Adafruit ItsyBitsy M0 breakout board (Adafruit Industries, New York, NY, USA). The resolution of the ADC was set to 12 bits. An external reference voltage of 1.2 V was used. A resistor divider was used to convert the value of the measured signal to a value below the reference voltage. A control program was developed for the SAMD21 microcontroller to measure the output signal from the ionization sensor and for data logging on a PC. The statistical analysis used an advanced data analysis software package originally developed by StatSoft Inc.

### 2.2. Experimental Setup

The experimental setup is shown in Figure 3. The smoke detector, as well as the high-precision temperature and relative humidity sensor SHT85 (Sensirion AG, Stäfa, Switzerland)), were placed in a sealed chamber with a mixing fan. The SHT85 sensor was connected to an Arduino Uno microcontroller (Arduino, Turin, Italy) and the data recorded on a PC. The chamber was equipped with four tubing ports. Two of the ports were used for the sampling and outlet lines of the MiniWRAS 1371 (Grimm Aerosol Technik GmbH, Ainring, Germany) aerosol spectrometer, which was used as a reference device [40]. The other two ports were used as an inlet and outlet for the HEPA filtered air or test aerosol. A spark generator GFG1000 (Palas GmbH, Karsluhe, Germany) equipped with graphite electrodes was used for generating the test aerosol [41]. Solid particles generated by a GFG1000 generator tend to form aggregates and agglomerates. The MiniWRAS 1371 was selected as a reference device due to its broad particle size range from 10 nm to 35 µm.

### 2.3. Methods

Before the experiment, the test chamber was flushed for 30 min with HEPA-filtered, dehumidified air sourced from the compressed air system. The states of the valves at various stages of the experiment are shown in Table 1.

Several tests were conducted to assess the possibility of determining the nanoparticle number concentration in the aerosol. The output signal stability from the smoke detector was assessed in a sealed chamber under constant temperature and humidity conditions. During the humidity effect test, the pump was turned on for 15 s at 5-min intervals. The bubbler was filled with demineralized water. For a single dose of test aerosol, the argon flow (5 L/min) was turned on for approximately 20 s. During this time, aerosol was generated for 15 s. The particles generated were under argon atmosphere and were not diluted with air. The sparking frequency between the graphite electrodes during the tests was set to 2 Hz.

## 3. Results and Discussion

### 3.1. Signal Stability

The signal stability test was conducted in a sealed test chamber under constant temperature and relative humidity conditions (Figure 4). The mean values of the temperature and relative humidity are 22.19 °C ± 0.02 °C and 37.52% ± 0.08%, respectively. The output signal from the smoke detector was registered for 1800 s with a frequency of 1 Hz. The mean value of the output signal is 387.55 mV with a standard deviation of 2.42 mV. The linear regression line calculated for this dataset has a slope coefficient of 3.344 × 10^−4^. One minute was the smallest common time scale for measurements using the tested sensor and the reference device. In this case, the standard deviation is 0.63 mV, while the slope coefficient of the linear regression line is 2.024 × 10^−2^ (Figure 5).

### 3.2. Humidity Effect Test

Points corresponding to periods of air humidification were removed from the registered dataset based on the value of the standard deviation. Changes in the temperature and humidity during the test are shown in Figure 6. The mean value of the temperature is 21.55 °C ± 0.32 °C. In Figure 6b, it can be seen that the relative humidity decreases between successive additions of moisture. This is due to the presence of a diffusion dryer at the inlet of the MiniWRAS spectrometer; the air at the device outlet has a lower moisture content than air sampled from the test chamber. The mean value of the output signal is 386.820 mV with a standard deviation of 0.852 mV. Figure 7a,b show the relationship between the output signal and the relative humidity and temperature of the air. The linear regression line calculated for the relative humidity dataset has a slope coefficient of −1.214 × 10^−4^; thus, it can be concluded that the value of the output signal was constant during the experiment. The air temperature dependence was approximated by a second-degree curve, where the slope coefficient was −8.113 × 10^−2^.

### 3.3. Argon Effect Test

Changes in the temperature and humidity during the argon effect test are shown in Figure 8a. The vertical lines indicate the moments at which argon was added to the test chamber. The mean values of the temperature and relative humidity are 20.68 °C ± 0.14 °C and 16.76 ± 0.41%, respectively. It can be seen in Figure 8b that each dose of argon causes a decrease in relative humidity, although both temperature and humidity increased during the test. It can be assumed that the slight increase in humidity is due to evaporation of moisture from the walls of the chamber. The aerosol concentration measured in the test chamber using the reference counter is below its lower detection limit, set at 3000 particles/cm^3^ (Figure 9a). The lower detection limit for 100 nm particles is estimated at the level of 15,000 particles/cm^−^^3^. In the case of the smoke detector, each dose of argon increased the value of the output signal (Figure 9b). The mean increase is 2.003 mV ± 0.372 mV per dose.

### 3.4. Effect of Aerosol under Constant Humidity Conditions

In this experiment, measurements with an aerosol concentration below the MiniWRAS lower detection limit were removed from the recorded dataset. Changes in the temperature and humidity during the test are shown in Figure 10. The vertical lines indicate the moments at which test aerosol was added to the test chamber. The mean values of the temperature and relative humidity are 21.82 °C ± 0.22 °C and 2.56 ± 0.15%, respectively. It can be seen in Figure 10b that each dose of aerosol causes a permanent increase in temperature by about 0.25 °C and a temporary increase in relative humidity by about 0.5%. The size distribution of the generated soot aerosol, calculated from data points where the total particle number concentration exceeded 3000 particles/cm^3^, is shown in Figure 11. It is a log-normal distribution with a number-weighted mean particle diameter of 50.35 nm. The share of particles smaller than 100 nm is 97.89%.

The comparison of the readings from the reference device and the smoke detector is shown in Figure 12. For the output signal from the smoke detector, the effect of adding argon was subtracted for each dose of aerosol. It can be seen that the aerosol particle number concentration drops from the initial value of about 35,000 particles/cm^3^ to a level below the lower limit of detection for MiniWRAS in about 10 min after each dose. This is because most of the particles in the aerosol stream reaching the MiniWRAS are lost in the measurement process (they are captured on the deposition electrode in front of the Faraday cup electrometer). Moreover, it can be seen that the output signal from the tested sensor increases with a delay as compared with the MiniWRAS. This is due to the fact that, while there is a mixing fan in the test chamber, there is no forced flow through the smoke detector’s ionization chamber.

The linear regression model between the readings from the MiniWRAS and the smoke detector is shown in Figure 13. The slope coefficient is 2747, while the coefficient of determination is 0.893. The aerosol concentration change per 1 mV change of the output signal for the calculated regression model is 2747.27 particles/cm^3^. The tested sensor by Dahl et al. [38] showed a linear response for concentration change of KCl calibration aerosol, candle smoke, and welding fumes.

### 3.5. Effect of Aerosol under Variable Humidity Conditions

As before, the measurements with the aerosol concentration below the MiniWRAS lower limit of detection were removed from the recorded dataset. The vertical lines in Figure 14 indicate the moments at which aerosol was added to the test chamber. The air was humidified before the addition of the individual aerosol doses. Changes in the temperature and humidity during the test are shown in Figure 14. The mean value of the temperature is 21.62 °C ± 0.33 °C.

The size distribution of the test aerosol, calculated from all data points where the total concentration exceeded 3000 particles/cm^3^, is shown in Figure 15. It can be seen that the distribution differs from that shown in Figure 10. There are more particles in the size distribution range below the most numerous fraction, which is the same in both cases. This is likely due to the accumulation of the smallest particles from consecutive doses. In this case, the number-weighted mean particle diameter is 39.98 nm. The share of particles smaller than 100 nm is 98.53%.

The comparison of the readings from the reference device and the smoke detector is shown in Figure 16. For the output signal from the smoke detector, the effect of adding argon was subtracted for each dose of aerosol. As shown in Figure 16b, with each successive dose of aerosol, the value of the output signal recorded for the lowest aerosol concentrations increases. The linear regression model between the readings from the MiniWRAS and the smoke detector is shown in Figure 17. The slope coefficient is 1729, while the coefficient of determination is 0.806; therefore, both coefficients are lower than in dry conditions. The aerosol concentration change per 1 mV change of the output signal for the calculated regression model is 1728.60 particles/cm^3^. This is a lower value as compared with that determined for the dry conditions.

## 4. Conclusions

This research on the response of an ionization detector to changes in environmental parameters was carried out in controlled laboratory conditions. The detector was installed on a test stand with gas flow velocity control, an aerosol sampling probe, and an integrated temperature, relative humidity, and air pressure sensor. The research results confirm that an ionization detector can be used to measure the concentration of nanoaerosols. The tests with an aerosol of synthetic carbon black particles demonstrated that the value of the output signal from the detector showed linear dependence with the concentration of particles.

The results showed that the modified smoke detector could be used to detect soot particles smaller than 100 nm. The percentages of particles smaller than 100 nm were 97.89% (test under constant humidity conditions) and 98.53% (test under variable humidity conditions). The output signal from the device varied linearly in the aerosol concentration range from 3000 particles/cm^3^ to 35,000 particles/cm^3^. The dependence of this signal on relative air humidity was also linear, while the dependence on air temperature was approximated by a second-degree curve. The linear regression line calculated for the relative humidity dataset had a slope coefficient of −1.214 × 10^−4^. The dependence on air temperature was approximated by a second-degree curve, with a slope coefficient of −8.113 x 10^−2^. In each case, the obtained relationships were characterized by a high correlation coefficient R^2^. Air humidity affected the aerosol concentration measurement results, which may be related to modifications of the nanoparticle surface. This research shows that the main limitations of the described method for nanoparticle detection may be an aerosol concentration above 30,000 particles/cm^3^ and relative air humidity outside the range of 40–60%. It also seems likely that the detection of nanoparticles is reduced at low air temperatures.

The World Health Organization (WHO) recommendations have adopted threshold values for nanomaterials based on the numerical concentration and specific density of nano-objects and their agglomerates and aggregates (NOAA) proposed by the Sociaal-Economische Raad (SER), an advisory body of the Dutch government. These are the limit values, and exceeding these limits should result in the application of appropriate exposure control measures. The threshold for metallic particles is 20,000 particles/cm^3^, and for metal oxides, carbon black, fullerenes, dendrimers, polystyrene, etc., it is 40,000 particles/cm^3^. At workplaces where nano-objects are used as substrates or where there is a risk of their spontaneous formation as a result of failures or processes, devices monitoring the concentration of nanoaerosols should be used. However, devices currently available on the market based on the principle of mobility measurement in an electric field are so expensive that this limits their use to periodically performed control measurements. The proposed solution, thanks to the use of ready-made components and existing technologies, makes it possible to construct a measuring device that is significantly cheaper. The ionization detector could be used for continuous monitoring of the presence of nano-object aerosols at workplaces. Future work is now ongoing in our laboratories on the development of two variants of a nanoparticle sensor using calibration curves, taking into consideration the influence of temperature and relative humidity of air and with a reference measurement carried out for filtered air.

## Figures and Tables

**Figure 1 sensors-22-03639-f001:**
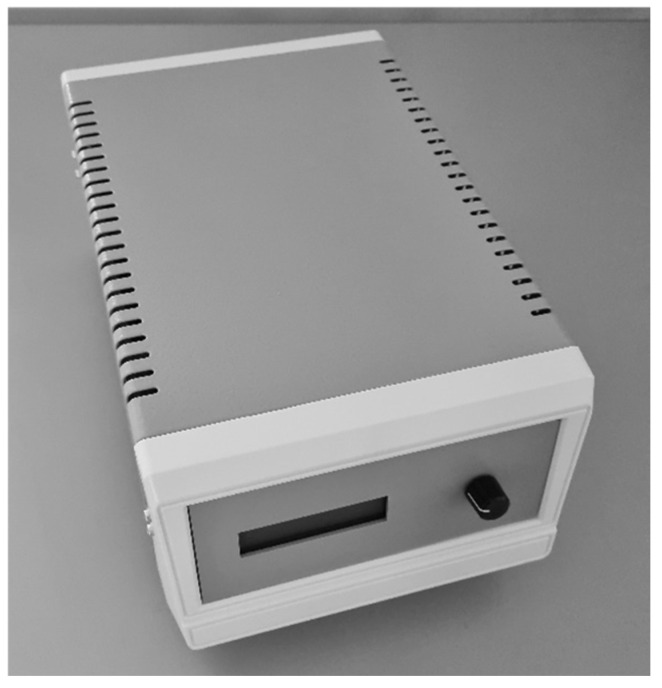
The model nanoparticle detector.

**Figure 2 sensors-22-03639-f002:**
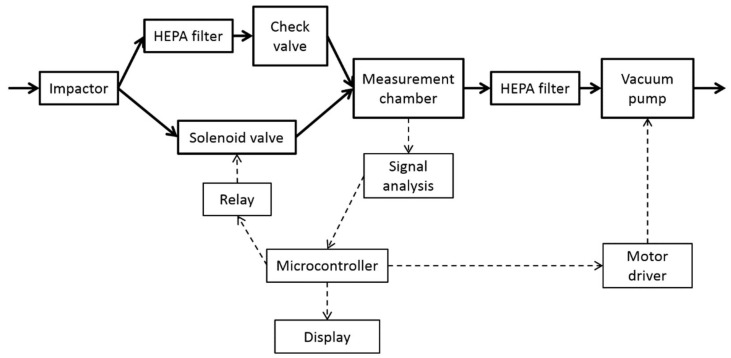
Block diagram of the designed nanoparticle detector.

**Figure 3 sensors-22-03639-f003:**
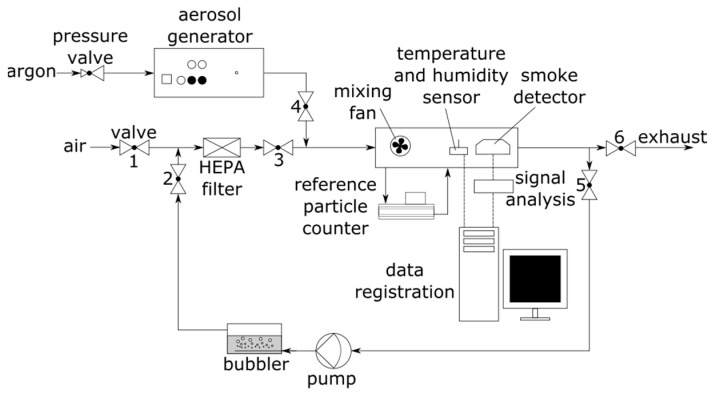
Experimental setup.

**Figure 4 sensors-22-03639-f004:**
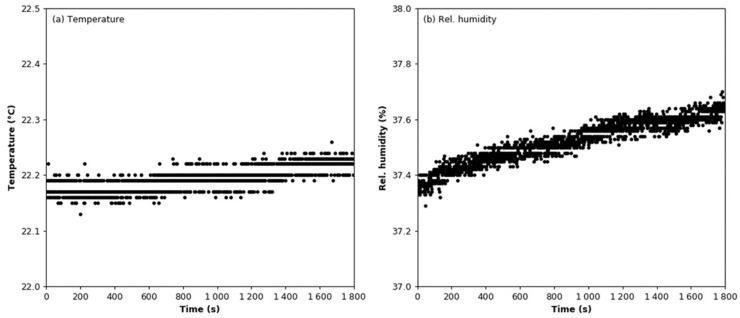
Environmental conditions during the stability test: (**a**) Air temperature; (**b**) relative humidity.

**Figure 5 sensors-22-03639-f005:**
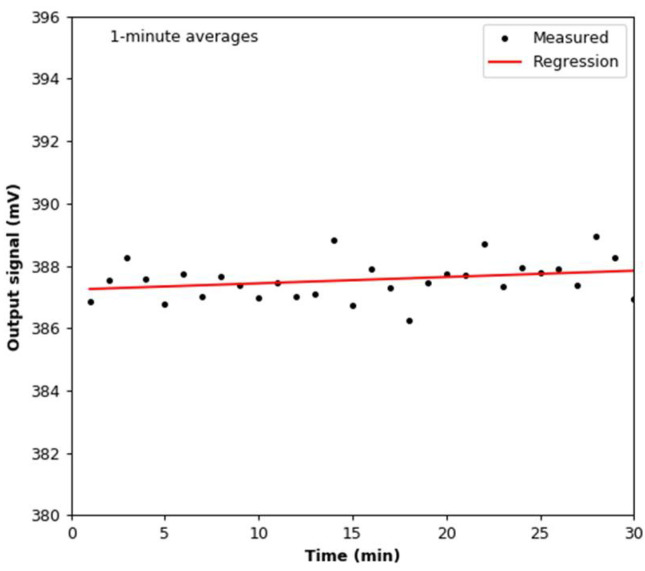
Variations in the output signal during the stability test: 1-**min** averages.

**Figure 6 sensors-22-03639-f006:**
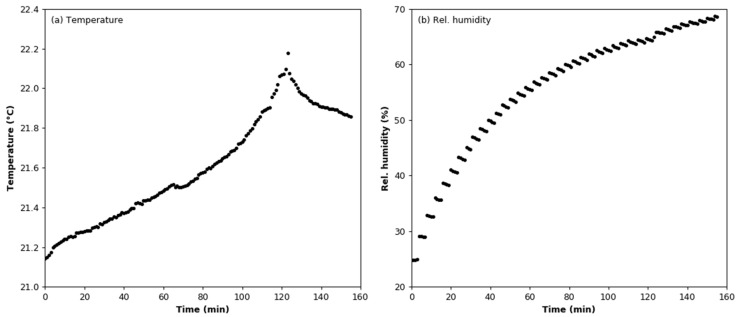
Environmental conditions during the humidity effect test: (**a**) air temperature; (**b**) relative humidity.

**Figure 7 sensors-22-03639-f007:**
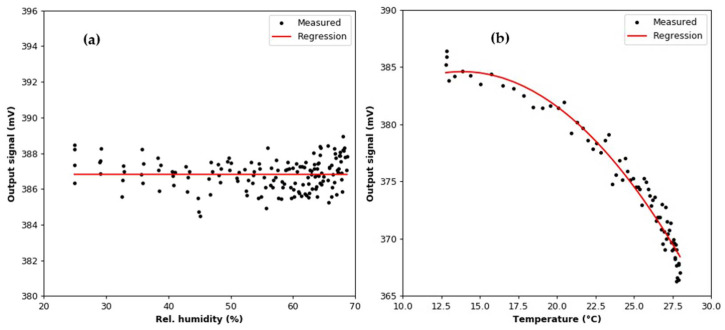
Variation in output signal during the (**a**) humidity and (**b**) temperature effect tests.

**Figure 8 sensors-22-03639-f008:**
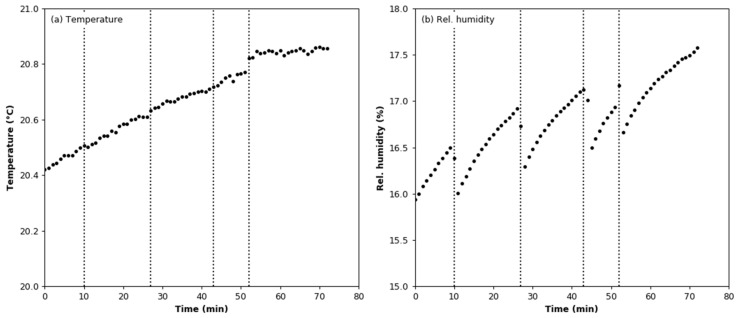
Environmental conditions during the argon effect test: (**a**) air temperature; (**b**) relative humidity.

**Figure 9 sensors-22-03639-f009:**
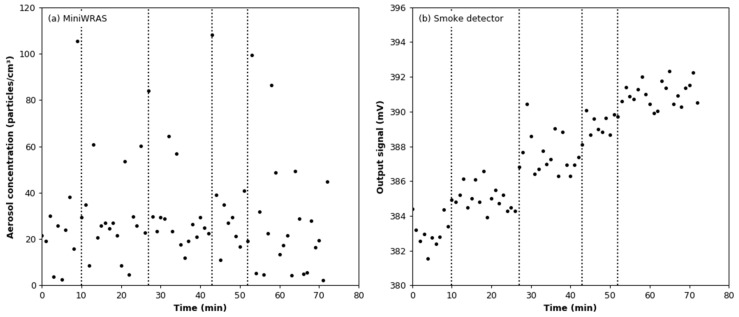
Readings from aerosol detectors during argon additions: (**a**) total number concentrations measured by the MiniWRAS; (**b**) output signal from the smoke detector.

**Figure 10 sensors-22-03639-f010:**
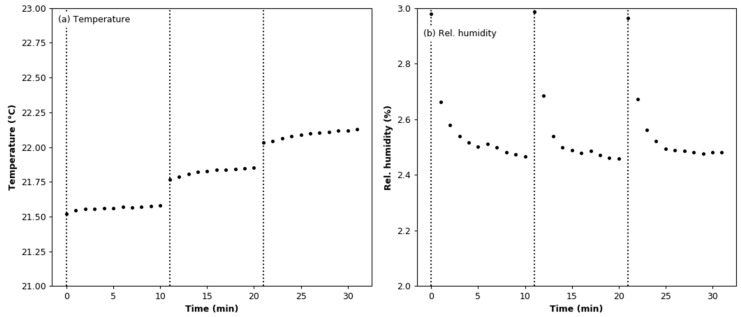
Environmental conditions during the soot effect test: (**a**) air temperature; (**b**) relative humidity.

**Figure 11 sensors-22-03639-f011:**
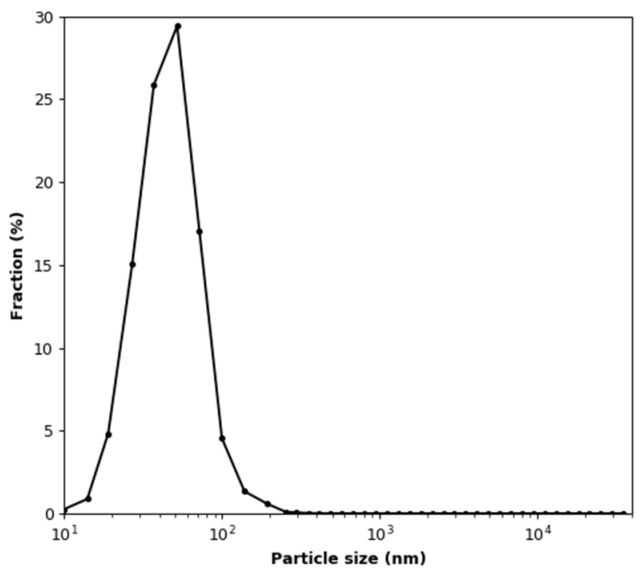
Test aerosol size distribution during the soot effect test.

**Figure 12 sensors-22-03639-f012:**
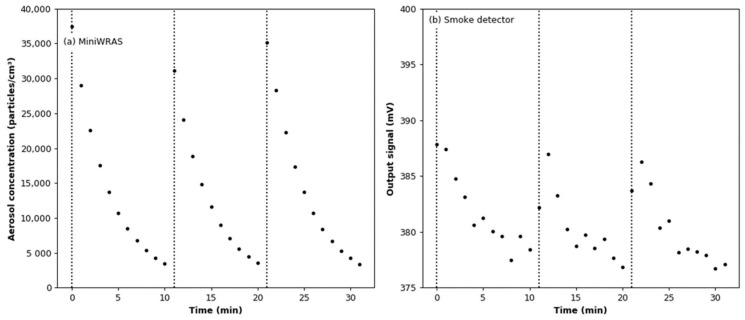
Readings from the aerosol detectors during the soot effect test: (**a**) total particle number concentrations measured by the MiniWRAS; (**b**) output signal from the smoke detector.

**Figure 13 sensors-22-03639-f013:**
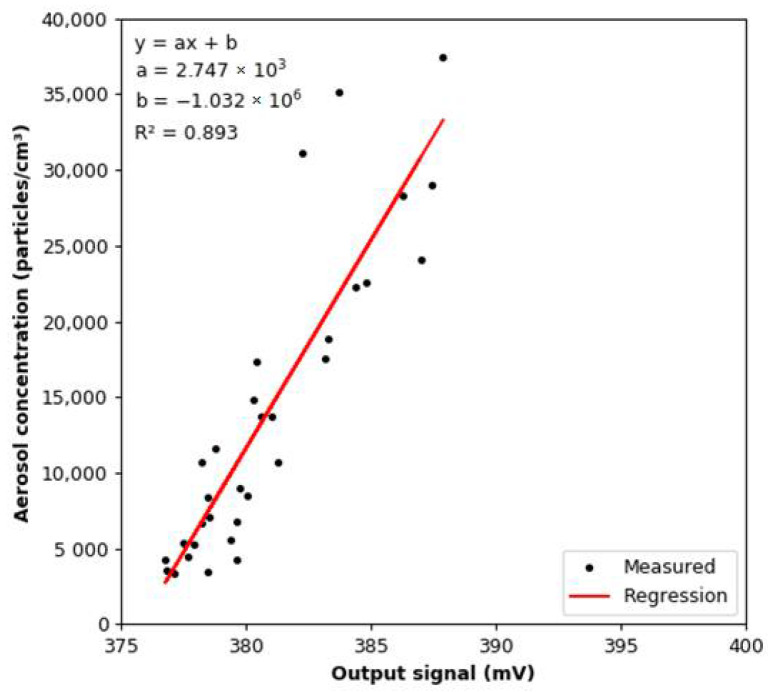
Correlation between the total aerosol concentration and the value of the output signal from the smoke detector under constant humidity conditions.

**Figure 14 sensors-22-03639-f014:**
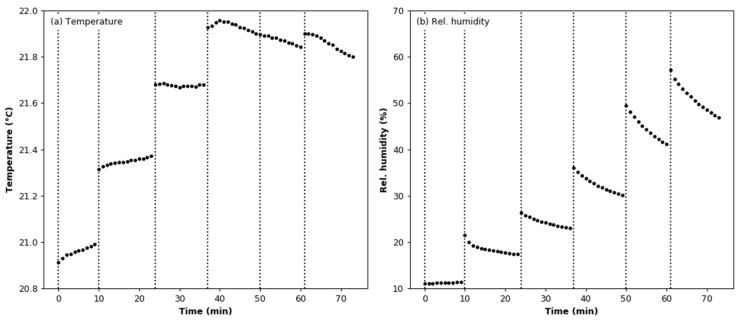
Environmental conditions during the soot effect test under variable humidity conditions: (**a**) air temperature; (**b**) relative humidity.

**Figure 15 sensors-22-03639-f015:**
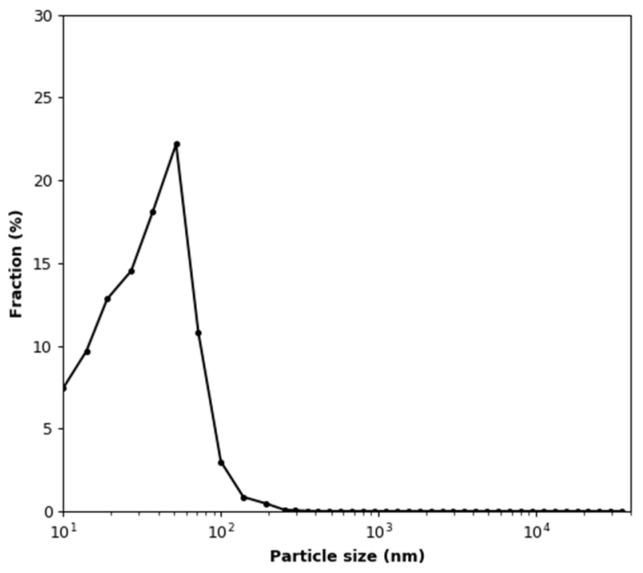
Test aerosol size distribution during the soot effect test under variable humidity conditions.

**Figure 16 sensors-22-03639-f016:**
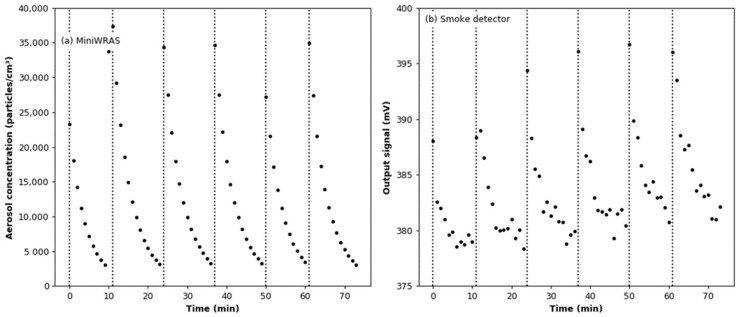
Readings from the aerosol detectors during the soot effect test under variable humidity conditions: (**a**) total particle number concentrations measured by the MiniWRAS; (**b**) output signal from the smoke detector.

**Figure 17 sensors-22-03639-f017:**
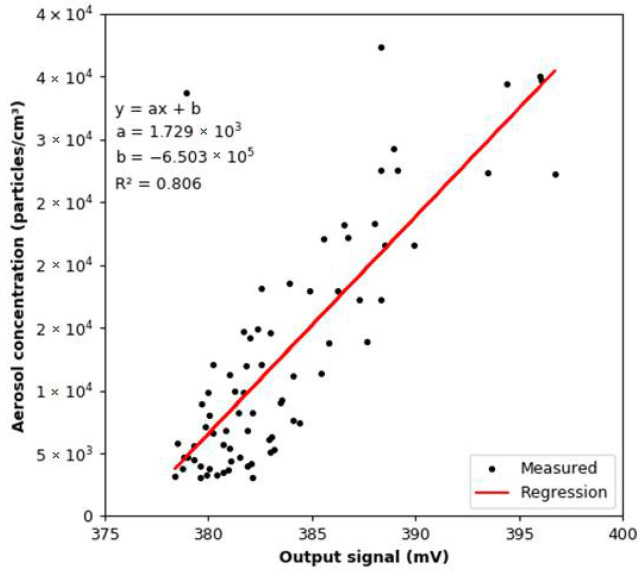
Correlation between the total aerosol concentration and the value of the output signal from the smoke detector under variable humidity conditions.

**Table 1 sensors-22-03639-t001:** The states of the valves in the experimental setup (Figure 3) at various stages of the experiment.

Valve	Air Preparation Stage (Air Flushing of the Chamber)	Aerosol Generating Stage (Adding Test Aerosol, Change in Aerosol Concentration)	Air Humidification Stage (Change in Air Humidity)
**1**	opened	closed	closed
**2**	closed	closed	opened
**3**	closed	closed	closed
**4**	closed	opened	closed
**5**	closed	closed	opened
**6**	opened	opened	closed

## Data Availability

Data available on request from the corresponding author.

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
