# Peer review of "The Influence of Air Humidity on the Output Signal from an Ionization Smoke Detector in the Presence of Soot Nanoparticles"

_sensors, 2022, doi:10.3390/s22103639_

Round 1

Reviewer 1 Report

This is a good study that characterizes a low-cost sensor for nano-particles. The need for the experiment is explained well and the rationalization is sound. The manuscript makes a good link to the health effects of these particles and notes that tradition sensors are very expensive.

I have some relatively minor comments that should be address:

Line 10: define NOAA/don’t use abbreviation in the abstract.
Line 64: remove “huge”.
Lines 70-72: give citation(s) for this global trend.
Lines 78-80: The structure of the sentence is confusing and the meaning is not clear.
Line 97: remove stray line break.
Lines 98-99: Explain why the unity amplifier is necessary. My guess is for the high impedance, but make explicit.
Line 135: Explain what the sparking frequency means.
Figures 12 & 13: this is a very good fit, but the three highest points are off the line a bit (Fig 13). Looking at Figure 12, this is due to the time synchronization issue discussed at Lines 217-220. Try removing these three points and see how the linear fit changes.
Figure 18: need a bit more explanation at lines 271-279 to explain how this graph was produced. Also, it would be useful to consider doing one linear model with both humidity and particle density. I believe each fit is done separately for one size distribution? Need more information.
Line 296: where is the data that shows the second-degree curve for the temperature effect?

Author Response

We are grateful for the detailed review of our paper, for the opinion on our work and suggestions that allowed us to improve and enhance it. Your comments were very useful for the editorial work on the revised version of our paper.

In reply to the Report of the Referee #1, we have introduced the attached changes.

Reviewer 2 Report

Manuscript ID: sensors-1653614. Influence of air humidity on the output signal from an ionization smoke detector in the presence of soot nanoparticles.  

The main objective of this study is to analyze the influence of air humidity on the output signal of an ionization smoke detector in the presence of soot nanoparticles.

Comments:

  1. The keyword "environmental engineering" is very general. Please check.
  2. The occupational safety and health keyword is not in line with the main theme of the article. Please check.
  3.  Please include quantitative information to support the findings shown in the abstract.
  4. Please clearly visualize in the abstract the objective of the study.
  5. Please avoid short paragraphs in the introduction. These must have an integral development. For example: L29-32, L33-36, L49-51, L82-83etc. Some paragraphs could be integrated.
  6. Significantly improve the introduction of the manuscript. Clearly visualize the objective of this study and its practical usefulness.
  7. I suggest moving Figure 1 to the chapter on materials and methods.
  8. Please include in the introduction information on the effect of climatic variables on the use of this type of detectors.
  9. Please include subtitles in the chapter on materials and methods to better organize the information displayed. For example, a section with the technical characteristics of the detector would give greater clarity.
  10. Assess the practical usefulness of Figure 2. It is probably better to detail Figure 1 and include it here.
  11. Please include a photograph of the detector in the materials and methods chapter.
  12. In the chapter on materials and methods there are also very short paragraphs. Please check.
  13. Table 1 requires more technical details to be relevant in this paper.
  14. Please remove the extra space in L130.
  15. The chapter on materials and methods should be significantly improved. More technical detail is needed. For example, detail the information collection system, expose in detail the analysis of information, and the statistical tests used.
  16. Was specialized software used in this research? Include it in materials and methods.
  17. Please include supporting references in the materials and methods chapter.
  18. Figure 4 would have a better display if it were in color.
  19. I suggest that all figures on the paper be in color.
  20. In the chapter on results and discussion there are also short paragraphs. Discuss the results in greater technical depth. Additionally, contrast their findings with other researchers. Include the references used.
  21. The manuscript has too many figures. Some of them could be integrated and presented in the same figure.
  22. Include an abbreviation section.
  23. Please include the main limitations of this study.
  24. Significantly improve conclusions. I suggest supporting them with quantitative information resulting from this research. Additionally, include the practical usefulness of this research.
  25. This study is relevant due to the development of novel technologies for the detection of contaminants. However, paper presents problems of structuring according to the standards of scientific journals. With a significant improvement, from the comments made, it is quite possible to recommend its publication.

Author Response

We are grateful for the detailed review of our paper, for the opinion on our work and suggestions that allowed us to improve and enhance it. Your comments were very useful for the editorial work on the revised version of our paper.

We will commission MDPI the editing of the English language and the required style

In reply to the Report of the Referee #2, we have introduced the attached changes.

Round 2

Reviewer 2 Report

The authors significantly improved the article based on the comments made. Therefore, I suggest its publication in the journal.